# Community determinants of COPD exacerbations in elderly patients in Lodz province, Poland: a retrospective observational Big Data cohort study

Anna Kowalczyk,[1] Katarzyna Kosiek,[2] Maciek Godycki-Cwirko,[1] Izabela Zakowska [1]

¹Centre for Family and Community Medicine, Faculty of Medical Sciences, Medical University of Lodz, Lodz, Poland
²Family Doctors' Clinic, Lodz, Poland

**Correspondence to**
Dr Izabela Zakowska;
izabela.zakowska@umed.lodz.pl

## ABSTRACT

**Objectives** To evaluate the prevalence and identify demographic, economic and environmental local community determinants of chronic obstructive pulmonary disease (COPD) exacerbations in elderly in primary care using Big Data approach.

**Design** Retrospective observational case–control study based on Big Data from the National Health Found, Tax Office and National Statistics Center databases in 2016.

**Setting** Primary care clinics in the Lodz province in Poland.

**Participants** 472 314 patients aged 65 and older in primary care, including 17 240 patients with COPD and 1784 with exacerbations (including deaths).

**Outcome measures** Exacerbations with demographic, economic and environmental local community determinants were retrieved. Conditional logistic regression for matched pairs was used to evaluate the local community determinants of COPD exacerbations among patients with COPD.

**Results** The overall prevalence of COPD in the population of elderly patients registered in primary healthcare clinic clinics in Lodz province in 2016 was 3.65%, 95% CI (3.60% to 3.70%) and the prevalence of exacerbations was 10.35%, 95% CI (9.89% to 10.80%). The high number of consultations in primary care clinics was associated with higher risk of COPD exacerbations (p=0.0687). High-income patients were less likely to have exacerbations than low-income patients (high vs low OR 0.601, 95% CI (0.385 to 0.939)). The specialisation of the primary care physician did not have an effect on exacerbations (OR 1.076, 95% CI (0.920 to 1.257)). Neither the forest cover per gmina (high vs low OR 0.897, 95% CI (0.605 to 1.331); medium vs low OR 0.925, 95% CI (0.648 to 1.322)), nor location of gmina (urban vs urban–rural OR 1.044; 95% CI (0.673 to 1.620), (rural vs urban–rural OR 0.897, 95% CI (0.630 to 1.277)) appears to influence COPD exacerbations.

**Conclusions** Big Data statistical analysis facilitated the evaluation of the prevalence and determinants of COPD exacerbation in the elderly residents of Lodz province, Poland.

Modification of identified local community determinants may potentially decrease the number of exacerbations in elderly patients with COPD.

## STRENGTHS AND LIMITATIONS OF THIS STUDY

⇒ Retrieving knowledge from three relevant community big databases (National Health Found, Tax Office and National Statistics Centre).
⇒ Evaluation of prevalence of chronic obstructive pulmonary disease (COPD) and determinants of COPD exacerbations in elderly primary care patients in Lodz province in Poland.
⇒ Its main limitation was the lack of standardisation in the method of recording exacerbations between primary care, accident and emergency departments and hospital wards.
⇒ No data existed for some parameters, such as air quality for gaseous and dust pollutants, at the commune level for the examined period.

## INTRODUCTION

Chronic obstructive pulmonary disease (COPD) is the third-leading cause of death worldwide, causing 3.23 million deaths in 2019.[1] Despite the great effort directed at prevention and treatment guidelines, the prevalence continues to increase and is now estimated to range from 4% to 10% worldwide; this value varies considerably across the European Union, that is, from 1.26% to 13.87%, partly because of the differences in the definitions used.[2–4] In Poland, a medium-sized country with a population of about 38 million inhabitants, studies suggest a COPD prevalence of 11% in the cities of Warsaw and Lublin; however, data are scarce in other regions.[5]

COPD is a progressive disease characterised by exacerbations, that is, an acute worsening of the respiratory symptoms, which may lead to hospitalisation.[6–8] In addition, patients with stable COPD often have acute exacerbations (AECOPD) due to risk factors, comorbidities, poor treatment adherence and insufficient treatment.[9] Frequent exacerbations, defined as at least two episodes per year, are

considered a marker of increased burden and reduced survival in clinical epidemiology studies.[10]

Before such exacerbations can be effectively prevented, there is a need to identify the factors contributing to them. To date, most studies have focused on factors specific to patients with COPD themselves, rather than on external factors.[11 12] However, it is possible that socioeconomic status (SES) may play a role. SES is a complex assessment of a person's economic and social position in society based on several factors including income, occupation, home and neighbourhood environment, and educational attainment.[13]

COPD appears to have a stronger socioeconomic element than other common causes of morbidity and death, including cardiac disease.[13] Also, low income appears to have a greater influence on the prevalence of COPD than asthma.[14] In Canada, where the overall prevalence of COPD was 5.69%, this prevalence was found to be significantly higher among older patients, and those who are currently smoking, underweight or obese, and those with total personal income <US$20 000 and some postsecondary education.[15]

A number of environmental factors can have effects on health, both positive and negative.

Green spaces, parks and community gardens support community health by reducing stress, promoting physical activity, and improving perceived general health.[16] These measures can be identified and modified to promote positive health behaviours and avoid negative ones,[17] and to shape specific promotional and educational initiatives aimed at reducing the prevalence of COPD and its exacerbations.

Recently, Big Data analyses have made it possible to measure, analyse, and document the health of individuals and populations, with the aim of improving public health and enhancing decision-making at the individual and community levels.[18] Such applications have been used to monitor the quality of medical and healthcare facilities, improve patient service, detect disease spread earlier, provide new insights into disease mechanisms through risk factors and improve treatments.[19 20]

Earlier studies on COPD exacerbations have been mostly prospective and have involved patients from secondary and/or tertiary care. In contrast, this study examines associations between well-known patient-level risk factors and triggers of exacerbations of COPD, including local community factors.

## Objectives

To estimate the prevalence of COPD in elderly patients in the province of Lodz Poland, and to determine the prevalence of AECOPD in this population.

To identify local community determinants of disease exacerbations among elderly patients with COPD.

## METHODS

A retrospective observational case–control cross-sectional study was performed among elderly patients with COPD in the Lodz province in Poland. The study was conducted between 1 January 2016 and 31 December 2016.

### Data sources

Data were retrieved from three Big Data databases of the National Health Fund (NFZ) (National Health Service), US (Tax Office) and GUS (Central Statistical Office).

The NFZ electronic health records (EHRs) database was used to gain information on sociodemographic (age, sex), inpatient and outpatient healthcare utilisation, including costs of hospitalisation, and the cost of dispensed medications in the form of anatomical therapeutic chemical (ATC) codes, as well as other corresponding costs and information on physicians working in primary healthcare clinics (PHC) in the study area.

The US database provided information on the income of residents aged 65 and older (65+) in the province of Lodz.

The GUS database contained variables potentially associated with exacerbations for 177 gminas (local communities) within the Lodz province.

Local community factors potentially associated with COPD exacerbations in the elderly were selected and classified by experts into (1) health factors, (2) socioeconomic factors and (3) community environmental factors at gmina level, according to the Remington and Catlin methodology.[21]

### Participants

All participants were patients registered in PHC clinics, aged 65+, residents of Lodz province. All were assigned to a particular gmina; it was assumed that the majority of patients aged 65+ choose a healthcare clinic close to their place of residence. COPD was indicated by the presence of the International Classification of Diseases (ICD-10) code J44 in patients' medical records; exacerbations were defined as cases 'hospitalised with the J44 code as a main reason for admission'.

### Big data quality process

The Big Data quality process was carried out according to the SAS Data Management Methodology. After cleaning the data, the rules and alerts for monitoring were defined (details given in online supplemental appendix).

### Statistical analysis

The data were analysed as described previously in protocol.[22] In addition, the propensity score matching (PSM) technique and multivariate conditional logistic regression for matched pairs were applied to estimate the local community determinants of AECOPD.[23]

Descriptive statistics were calculated for the total cohort of patients with aged 65+ before and after PSM. Missing data were excluded from the analysis. The relationships between tested variables and exacerbations (1-with, 0-without) were determined using univariate high-performance logistic regression models, and univariate logistic regression models applied and stratified logistic regression (a single model is for each line). Continuous variables were categorised by quartile. Pearson's

**Table 1** Descriptive characteristics of COPD exacerbations in cohort of elderly patients with COPD (Lodz Province, 2016; case–control) after PSM

| Patient characteristic (after PSM) | Patients with COPD without exacerbations control (n=1492 pairs) | Patients with COPD with exacerbations cases (n=1492 pairs) | Total (n=2984) | OR (95% CI) Univariate analysis* |
|---|---|---|---|---|
| Patient level | | | | |
| Demographic | | | | |
| Patients with COPD (sum) (%) total | 1492 (50) | 1492 (50) | 2984 (100) | |
| Sex (strata variable) | | | | |
| Male | 886 (59.38) | 886 (59.38) | 1772 (59.38) | 1 Ref. Cat† |
| Female | 606 (40.62) | 606 (40.62) | 1212 (40.62) | 1.000 (0.864 to 1 .157)† |
| Age (years) (strata variable) | | | | |
| <75–85 | 600 (40.21) | 600 (40.21) | 1200 (40.21) | 1.000 (0.788 to 1.269)† |
| <85+> | 175 (11.73) | 175 (11.73) | 350 (11.73) | 1 Ref. cat†. |
| <65–75 | 717 (48.06) | 717 (48.06) | 1434 (48.06) | 1.000 (0.792 to 1.263)† |
| Patient's place of residence in gmina (localisation) | | | | |
| 1-urban vs 3 | 862 (57.77) | 864 (57.91) | 1726 (57.84) | 1.028 (0.848 to 1.246) |
| 2-rural vs 3 | 353 (23.66) | 358 (23.99) | 711 (23.83) | 1.04 (0.833 to 1.3) |
| 3-urban–rural (Ref. Cat.) | 277 (18.57) | 270 (18.1) | 547 (18.33) | 1 Ref. Cat. |
| Healthcare use | | | | |
| Specialisation of physician working in PHC | | | | |
| Family doctors vs other | 562 (37.67) | 587 (39.34) | 1149 (38.51) | 1.080 (0.926 to 1.259) |
| Other (Ref. Cat.) | 930 (62.33) | 905 (60.66) | 1835 (61.49) | 1 Ref. Cat. |
| No of consultations in PHC† | | | | |
| High (>12) vs low (<5) | 558 (37.4) | 689 (46.18) | 1247 (41.79) | 1.269 (0.982 to 1.639) |
| Medium <5–12> vs low (<5) | 783 (52.48) | 649 (43.5) | 1432 (47.99) | 0.831 (0.651 to 1.061) |
| Low (<5) (Ref. Cat.) | 151 (10.12) | 154 (10.32) | 305 (10.22) | 1 Ref. Cat. |
| No of COPD consultations in PHC (binary)‡ | | | | |
| 0–1 | 764 (51.21) | 834 (55.90) | 1598 (53.55) | p=0.0096‡ |
| 2+ | 728 (48.79) | 658 (44.10) | 1386 (46.45) | |
| Gmina/commune, postal codes level | | | | |
| Socioeconomic status | | | | |
| Total personal income of residents per PHC postal codes (tax office) | | | | |
| High (>21 056.52) vs low (<17 079.50) | 371 (24.87) | 345 (23.12) | 716 (23.99) | 0.696 (0.489 to 0.990) |
| Medium <17 079.50–21 056.52> vs low (<17 079.50) | 759 (50.87) | 768 (51.47) | 1527 (51.17) | 0.910 (0.722 to 1.147) |
| Low (<17 079.50) (Ref. Cat.) | 362 (24.26) | 379 (25.4) | 741 (24.83) | 1 Ref. Cat. |
| Environmental factors | | | | |
| Forest cover of the patient's place of residence in gmina (GUS) | | | | |
| High (>20.60%) vs low (<7.00%) | 348 (23.32) | 340 (22.79) | 688 (23.06) | 0.896 (0.648 to 1.24) |
| Medium <7.00%%–20.60%> vs low (<7.00%) | 776 (52.01) | 776 (52.01) | 1552 (52.01) | 0.94 (0.688 to 1.285) |
| Low (<7.00%) (ref. Cat.) | 368 (24.66) | 376 (25.2) | 744 (24.93) | 1 Ref. Cat. |
| Total | 1492 (50) | 1492 (50) | 2984 (100) | |

*The LOGISTIC procedure, conditional logistic regression for matched pairs.
†Sex and age were strata variables (the LOGISTIC procedure, logistic regression).
‡McNemar's test B/C ($\chi^2$=6.71, df=1, p=0.0096) in STATISTICA V.13.
cat, reference category; COPD, chronic obstructive pulmonary disease; PHC, primary healthcare clinic; PSM, propensity score matching.

correlation coefficient was estimated. Patients with exacerbations were matched retrospectively for gender and age to patients without exacerbations to ensure 1:1 group ratio without repetition using PSM. Patients who died were excluded from the analysis. Costs were calculated as the percentage of costs per patient to the mean COPD patient population and categorised by quartiles (online supplemental appendix).

After adjusting for potential covariates, the OR and CIs of exacerbations between cases and control patients were calculated by multivariate conditional logistic regression for matched pairs models. For the prevalence of the

disease, 95% CIs of the fraction of patients in the population were calculated.

Statistically significant findings were assumed as p<0.05.

The analysis was performed with the SAS V.9.4 statistical package, MLwiN V.2.24 and STATISTICA V.13.1.

## Study size/power calculation

The power was calculated for paired t-test for mean difference. Taking number of pairs 1492, calculated mean difference between number of consultations in PHC clinic for patients with exacerbation and without exacerbations was 1.214 (SD=9.658). The SD for the number of consultations in PHC clinic in group with exacerbation was 7.62 and in group without exacerbation was 6.76, the correlation between them was 0.103 (p<0.0001), which gave the power of 99.8%. A total of 499 pairs would allow 80% power.

## Patient and public involvement

This was a Big Data study using patients' anonymised EHRs without patient involvement.

## RESULTS
### Prevalence

The Lodz province comprised 2 485 323 inhabitants as of 31 December 2016.[24] According to the NFZ EHRs database within the Lodz province, 472 314 elderly patients were registered in PHC clinics with contracts with the NFZ (19% of total number of residents).

Prevalence of COPD in the population of elderly patients registered in PHC clinics in Lodz province in 2016 was 3.65% (95% CI (3.60% to 3.70%)) (17 240 patients coded as J44) of all patients (n=472 314). Among all patients with COPD, 10.35%, 95% CI (9.89% to 10.80%) (1784) were recorded as having an exacerbation; this group represents 0.38% of the entire elderly patient population in the Lodz province. During the study, 1 141 of 17 240 (6.62%) patients died and their data was excluded from further analysis. In total, 16 099 patients with COPD were included in the analysis.

### Characteristics of the COPD patient groups (raw data before PSM)

The group characteristic is presented in online supplemental table 1S and online supplemental table 1A.

In total, 1 492 (9.27%) of 16 099 patients with COPD reported exacerbations. The number of exacerbations per individual patient with COPD ranged from one to six. Most of these patients reported 1 exacerbation (1294 patients; 86.73%), with fewer reporting 2 (148 patients; 9.92%), 3e (32 patients; 2.14%), 4 (11 patients; 0.74%); 5 (5 patients; 0.34%) and 6 exacerbations (2 patients; 0.13%).

A statistically significant relationship was observed between sex and exacerbations (p<0.0001). 59.38% of patients in group with exacerbation were male. In addition, a significant relationship was noted between age and exacerbations (p<0.0001): the COPD group with exacerbations had a higher median age 75 years old (lower quartile=69, upper quartile=81) than the patients with COPD without exacerbations 73 years (q1=68, q3=80).

The number of COPD consultations in PHC clinics, classed as the number of healthcare consultations with the main diagnosis J44, ranged from 0 to 24 per patient during the study year, with the most common value being one consultation in PHC clinic per 65+ patient with COPD.

Among the people with exacerbation, 36.73% did not have any COPD-related consultations in the PHC clinic and 1.14% in the group without exacerbation. 19.17% patients in group with exacerbations had received one COPD consultation (52.72% in group without exacerbation), 12.8% two consultations (20.06% in group without exacerbation), 7.91% three consultations (10.36% in group without exacerbation), 23.39% four or more COPD consultations (15.72% in group without exacerbation).

The number of COPD consultations in 2016 was associated with the number of exacerbations (p<0.0001).

The number of consultations for any reason in PHC clinic per 65 +, patient with COPD during the study year ranged from 0 to 78, with the most common value being 10 consultations.

The number of consultations in PHC in 2016 was significantly associated with disease exacerbation (medium vs low p=0.0350 and high vs low p=0.0092). Among patients with exacerbation, about 46% had 12 or more consultations, 44% had a medium number of consultations (n=5–12) and only 10% hada low number (n=0–4).

In total, 2465 physicians were working in PHC clinics within the Lodz province, including 627 (25.44%) family doctors, and 1838 (74.56%) other physicians (general internal medicine, paediatrics, physicians without specialisation). Of the total physicians, 1674 (67.91%) were female. The median age of physicians was 52 (range 25–88) years.

Similar percentages of patients whose physician was a family doctor vs other were observed in the group of patient with COPD with exacerbations (39.34%) and in the group without exacerbations (37.54%) (p=0.1703).

There was an association between the numbers of packages of reimbursed drugs and exacerbations (R03 AC, AL, BA, BB, DA: p=0.0001, AK p=0.0015, R03DC p=0.4080), although these variables had n=1906 (11.84% missing data) and need to be carefully interpreted.

The COPD group with exacerbations demonstrated higher numbers of packages of reimbursed drugs with ATC codes R03 AC, AL, BA, BB, DA AK except R03DC compared with those without exacerbations, based on prescriptions provided to the patient.

The NFZ drug reimbursement costs per patient were significantly associated with exacerbations (p<0.0001, n=1906 (11.84%) of missing data). There are also a significant association between total stationary costs and exacerbations (p<0.0001) (online supplemental appendix). The total NFZ costs per patient (as sum of costs of PHC, ASC, hospitalisation, drug reimbursement and other) was

significantly associated with exacerbations (p<0.0001) (online supplemental appendix).

Income was also significantly associated with exacerbations (p<0.0001).

The group with exacerbations demonstrated lower income, by PLN 584.02 per year, compared with the group without exacerbations.

Forest cover of the gmina where the patient lived had a significant impact on the exacerbations (high vs low p=0.0014, medium vs low p=0<0.0001). Forty-three per cent of patients in group with exacerbations were residents of gminas with a lower level of forest cover.

COPD exacerbations were found to be significantly associated with patient's place of residence in gmina (urban, rural, urban–rural).

Patients with COPD exacerbations throughout the 1-year study period were matched for gender and age at baseline with patients without COPD exacerbations in a 1:1 ratio: n=1492 cases (50%) and n=1492 controls (50%) (before and after PSM: histograms in online supplemental figures 1–4).

The characteristics of patients with COPD with and without exacerbations after PSM are given in tables in online supplemental appendix table 2A and online supplemental table 2S.

Table 1 shows descriptive characteristics and local determinants univariate analysis of COPD exacerbations in cohort of elderly patients with COPD (Lodz province, 2016; case–control) after PSM.

PSM analysis identified similar relationships between variables and exacerbations as in the raw data. Median age was the same for patients with and without exacerbations (75 years, q1=69, q3=81, for both groups).

Patient consultations and consultations for COPD in PHC clinics had a weak association with exacerbations (high vs low p=0.0687).

High number of consultations in PHC clinics (any reason) had 46.18% of patients in group with exacerbations and 37.40% in group without exacerbations.

In addition, 36.73% of patients with exacerbations and 0.87% of patients without exacerbations had not received any COPD consultations in PHC clinic. When the number of consultations in PHC with the main diagnosis J44 increased from one to three, a decrease in the number of exacerbations was observed.

There was an association between the numbers of packages of reimbursed drugs and exacerbations (R03 AC, AL, BA, BB, DA: p=0.0001, AK p=0. 0078, R03DC p=0. 9022.

Patients with exacerbations used a greater number of prescriptions for reimbursed drug packages. Patients with exacerbations had higher numbers of prescriptions of reimbursed drugs from ATC groups with codes R03AC, AL, BA, BB, DA, AK compared with those without exacerbations. Again, the numbers of R03DC packages did not differ significantly between the groups.

Significant associations were found between total NFZ cost per patient (as the sum of costs of PHC, ASC, hospitalisation, drugs reimbursement and other) and COPD

exacerbations for both data before PSM (p<0.0001) (online supplemental appendix table 3A) and after PSM (p≤0.0001) (online supplemental appendix table 3B).

Of all patients in the exacerbations group, 37.53% had high total NFZ cost per patient, and 58.65% had medium. Among those without exacerbations, the costs were high in only 12%, medium in 41% and 46% had low costs.

A significant association was found between income and disease exacerbations (high vs low p=0.0439). There were fewer exacerbations in the group of patients with high income.

There was no association between the forest cover of the patient's place of residence in gmina and exacerbations (high vs low p=0.5082, medium vs low p=0.6986).

## Model of local determinants of COPD exacerbations for elderly patients with COPD

Table 2 shows local determinants of COPD exacerbations for 65+ patients. For more descriptive characteristics of the determinants, please see online supplemental material and appendix.

The final model includes five independent variables: number of consultations in PHC, income, physicians' specialisation in PHC, the area covered with forest in gmina of patient's residence and location of gmina (urban, rural, urban–rural).

Two variables seemed important: number of consultations in PHC (border significance) and income.

Patients with a medium number of consultations in PHC demonstrated a slightly lowerchance of exacerbations than those with a low number of consultations (medium <5–12> vs low (<5) OR=0.823, 95% CI (0.643 to 1.053)) (table 2).

Patients with COPD with a high number of consultations (>12) in PHC showed a trend for more exacerbations compared with those with a low number (<5) of consultations (OR 1.261, 95% CI 0.974 to 1.633). Patients with COPD living in areas with a high income had a lower chance of exacerbations than those in areas with a low income (high vs low OR=0.601 (0.385–0.939)). Living in a high-income area is a positive and protective factor vs living in a low income one.

Type of physician's specialisation in PHC had no influence on the exacerbations (family doctor vs other OR 1.076, 95% CI (0.920 to 1.257) neither the forest cover area (high vs low OR=0.897, 95% CI (0.605 to 1.331), medium vs low OR=0.925, 95% CI (0.648 to 1.322)) or patient's place of residence in gmina (urban vs urban–rural OR=0.897, 95% CI (0.630 to 1.277), rural vs urban–rural).

## DISCUSSION

Despite significant advances in the diagnosis and treatment of COPD, its incidence remains high. In addition, in some cases, the chance of exacerbation may increase as a result of local risk factors.

Fortunately, nowadays information on community health and healthcare system can be easily gathered from

**Table 2** Local determinants of COPD exacerbations in cohort of elderly patients with COPD (Lodz Province, 2016; case–control) after PSM

| Local determinants of COPD exacerbations (after PSM) | Final model conditional logistic regression for matched pairs OR (95% Wald confidence limits), (point estimate) |
|---|---|
| Patient level | |
| Demographic | |
| Patients with COPD (sum) (%) total | n=1492 pairs |
| Patient's place of residence in gmina (localisation) | |
| 1-urban vs 3 | 1.044 (0.673–1.620) |
| 2-rural vs 3 | 0.897 (0.630–1.277) |
| 3-urban–rural (Ref. Cat.) | 1 Ref. Cat. |
| Healthcare uses | |
| Specialisation of physician working in PHC | |
| Family doctors vs other | 1.076 (0.920–1.257) |
| Other (Ref. Cat.) | 1 Ref. Cat. |
| No of consultations in PHC | |
| High (>12) vs low (<5) | 1.261 (0.974–1.633) |
| Medium<5–12> vs low (<5) | 0.823 (0.643–1.053) |
| Low (<5) (Ref. Cat.) | 1 Ref. Cat. |
| Gmina/commune, postal codes level | |
| Socioeconomic status | |
| Total personal income of residents per PHC postal codes (tax office) | |
| High (>21 056,52) vs low (<17 079,50) | 0.601 (0.385–0.939) |
| Medium <17 079,50–21 056.52> vs low (<17 079,50) | 0.824 (0.581–1.169) |
| Low (<17 079,50) (Ref. Cat.) | 1 Ref. Cat. |
| Environmental factors | |
| Forest cover of the patient's place of residence in gmina (GUS) | |
| High (>20,60%) vs low (<7,00%) | 0.897 (0.605–1.331) |
| Medium <7.00%%–20,60%> vs low (<7,00%) | 0.925 (0.648–1.322) |
| low (<7,00%) (ref. Cat.) | 1 Ref. Cat. |

The LOGISTIC procedure, model binary logit, 1:1 matched case–control conditional logistic regression for matched pairs. Sex and age variables were included as stratification variables (the 1:1 matched case–control), thus they are not explicitly included in the conditional logistic regression for matched pairs models.
COPD, chronic obstructive pulmonary disease; PHC, primary healthcare clinic; PSM, propensity score matching.

a number of large databases, as Big Data and analysed statistically.[25]

Big Data technology can be used to improve the course of chronic disease and its exacerbations. For example, Korean researchers were able to match COPD exacerbations with external factors.[26 27]

Similarly, this study applied Big Data analysis to a large primary care cohort of 472 314 patients aged 65 years and older in the Lodz province, Poland. The results yielded new information on disease prevalence and its relationship with local community factors (demographic factors, healthcare use, economic and environmental factors).

A meta-analysis of COPD prevalence reported a pooled prevalence of 9.8% among men and 5.6% among women.[28]

In this study, this value was 3.65% among all elderly patients. COPD exacerbations, hospitalisations and deaths in women have been increasing in several countries, with more women being hospitalised for COPD in recent years[29]; indeed, the percentage of hospitalisations among women in Puglia, Italy was found to increase from 27.8% in 2001 to 35.6% in 2011.[30]

However, in our study 59.38% of patients with COPD with exacerbation were male (raw data p<0.0001).

More research is needed to verify these apparent gender differences, not only among hospitalised patients but also in their daily environment.

In a Danish study, the risk of hospitalisation for COPD increased with age until 80 years for females and 85 years for males.[31]

Our findings also indicate a significant relationship between age and exacerbation (raw data p<0.0001). Of those with exacerbations, 47% are aged 65–75 years, 40% are aged 75–85 years, and approximately 14% are 85 years of age or older. In our study, the age of patients in the exacerbation group was higher than in group without exacerbation: 75 years old vs 74 (raw data p<0.0001)

The high number of primary care consultations per patient and a low income in the local area were important determinants of exacerbations among patients with COPD aged 65+ (p=0.0784 border of significance and p=0.0253, respectively), whereas there was no effect of specialisation of the PHC physician, forest cover of the local area and the location of the gmina on rates of exacerbations.

In this study, patients with a higher number of consultations had a slightly higher risk of exacerbations compared with patients with a low number of consultations (OR 1.261; 95% CI 0.974 to 1.633).

It is likely that the former group have a greater chance of referral to a hospital by a primary care physician, and thus faster diagnosis and treatment.

Patients with high income are less likely to have COPD exacerbations than those with low income (OR 0.601; 95% CI 0.385 to 0.939). Other studies have reported similar trends.[32] This may be related to decreased access to healthcare services among lower income groups. Such problems with access may delay the diagnosis of COPD, leading to a poorer prognosis and increased morbidity and mortality.[33] Addressing this issue may reduce the workload of the healthcare system and allow more targeted use of available resources. Low income in the local area was an important determinant of exacerbations among patients with COPD aged 65+ and it can be explained by a number of other cofactors such as: more severe COPD, comorbidities, less inclination to use regular inhalers/medications, less healthy lifestyle or poor living conditions. The association between neighbourhood greenness and respiratory outcomes in adults appears ambiguous.[34] One cohort study including 1.3 million adults in Canada found that greater neighbourhood greenness could reduce the risk of respiratory mortality[35]; however, a Chinese cohort study concluded that neighbourhood greenness might in fact be a risk factor.[36]

In this study, forest cover per gmina was not found to have any statistically significant effect on COPD exacerbations.

Rural residence has been found to be associated with an increased incidence of COPD exacerbations, although not severe ones.[37] In countries with large rural populations, COPD is more prevalent in rural than urban areas. This disparity may be related partly to increased agricultural dust/aerosol exposure (wheat, grain, cotton dust, pesticides).[38] In addition, studies suggest an association with increased indoor pollution due to burning biomass fuels.[39]

In this study, patient's place of residence in gmina (urban, rural, urban–rural) did not appear to have any influence on the COPD exacerbations.

In addition, the specialisation of the physician in primary care did not have any effect on the COPD exacerbations.

Identification of patient population relied only on the ICD-10 J-44 code in patients primary care records, so patient with COPD in specialist clinics and on hospitals wards without this ICD-10 code might not be identified, what can be perceived as study limitation. The validation of the J-44 Code of the patients enrolled in the study was not possible at the PHC clinics level because the NFZ provided us with anonymised patient data.

Patients with code J44 were included in this study, without excluding codes for other diseases, including asthma, due to missing data in the variable 'Codes for comorbidities with COPD' (98.83% (466 780 out of 472 314) in total 65+ patients in PHC in Lodz province in 2016), so the variable was excluded from the analysis.

Another limitation of this study was lack of standardisation in the method of recording exacerbations between primary care, accident and emergency departments and hospital wards.

## CONCLUSIONS

This study investigates the determinants for AECOPD with the aim of developing activities to improve prevention and treatment interventions. Our findings regarding COPD in the Lodz province, Poland may also be used for formulating health policy recommendations by the NFZ and local government. The Big Data methodological approach used here can be applied to other chronic diseases. Further study is needed to determine the role of local medical and environmental factors in improving the health of patients with COPD version of the manuscript.

**Acknowledgements** We acknowledge the role of experts from various fields: primary care physicians, public health specialists, a Big Data analyst and data managers in public institutions (NFZ, GUS and US) in the study. The authors specially thank Edward Lowczowski for English language assistance.

**Contributors** The study design was conceptualised by AK, MG-C, KK and IZ. Analyses were performed by IZ, KK and IZ prepared the first draft of the manuscript. All authors read and agreed to the published version of the manuscript. IZ acts as guarantor of this manuscript.

**Funding** This work was supported by Narodowe Centrum Nauki (National Science Centre, Poland) grant number 2016/21/B/NZ7/02052.

**Competing interests** None declared.

**Patient and public involvement** Patients and/or the public were not involved in the design, or conduct, or reporting, or dissemination plans of this research.

**Patient consent for publication** Not applicable.

**Provenance and peer review** Not commissioned; externally peer reviewed.

**Data availability statement** Data are available on reasonable request. No additional data are available.

**ORCID iD**
Izabela Zakowska http://orcid.org/0000-0002-8085-7316

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
