## [Reviewer comments · BMJ Open]

ARTICLE DETAILS

TITLE (PROVISIONAL)	Community determinants of COPD exacerbations in elderly patients in Lodz province, Poland: a retrospective observational Big Data cohort study
AUTHORS	Kowalczyk, Anna; Kosiek, Katarzyna; Godycki-Cwirko, Maciek; Zakowska, Izabela

VERSION 1 – REVIEW

REVIEWER	Tariq, Syed Luton and Dunstable University Hospital, Respiratory Medicine
REVIEW RETURNED	01-Feb-2022

GENERAL COMMENTS	This study is a retrospective Big Data analysis to identify community-based factors associated with COPD exacerbations in the elderly population within a defined geographic area in Poland over a period of one year. After conversion of the data by applying propensity score matching, the study results are essentially negative, as apart from showing more exacerbations among the low income vs high income patients, there are no other significant results. The number of prescriptions/drugs, and the healthcare costs show statistical differences, but these are expected and likely reflect severity/GOLD class of COPD. All other community-based factors (specialty of primary care physician, forest cover, gender, number of PHC consultations, etc.) had little or no impact on rates of exacerbations. The association between the number of PHC consultations and hospital admissions may be explained simply by disease severity or even by an increased chance of being referred by the primary care physician to secondary care as these patients got seen more often by their primary care doctors. The study is retrospective, and such work has been done before. Hence, I suggest removing the word 'pioneering' from the first bullet point given in the 'Strengths and Limitations' section. There is no data on co-morbidities which could partly explain the differences in rates of exacerbation and hospitalization between the two study groups. I do not understand the abbreviations given for the packages of reimbursed drugs (pp 12 & 14; Tables). These need a clear explanation. The titles of tables 1 and 2 should be removed from the text. Both tables are too elaborate, and some of the data can be removed as it does not add to the analysis and results, such as the columns giving data on the total number of patients (n=16099) studied, and
---

	the mode (a descriptive statistic) for the number of PHC consultations. The breakdown of costs for the raw data and the data after PSM transformation deserves separate tables. There is no mileage in including the characteristics and specialties of primary care physicians in Table 1. These should simply be described in the text. Similarly, the details of patients who had died should also be removed from Table 1 and be explained in the text. The breakdown of age ranges at the end of Table 1 is not required and may simply be given in the text in 'Results' section. The 'Missing data' given in each of the 7 rows under the heading of 'Drugs/Number of Packages' may be removed, and simply marked with a symbol with an explanation given at the end of Table 1. The first histogram in Appendix on Big Data Quality section is difficult to follow. Do the darker portions of the bars represent patients who had exacerbations? The histograms on gender are redundant and may be removed. The labels, etc. for the remaining two histograms need to be translated from Polish into English. Though the references are appropriate, they have lots of errors. The months and dates of publications are not required and may be removed. The sources of funding, and the descriptions in parentheses as to the type of publication (Review, Meta-analysis, Editorial, etc.) are not needed as well. Please check the accuracy of the references as there are several minor errors, and some of them are incomplete. Ref 19 contains an erroneous title. The English needs improvement to make this manuscript more readable and acceptable. Although the statistical methods described are appropriate, I am unable to comment on the quality and outcomes of the analysis.
--	--

REVIEWER	MURO, Shigeo Nara Medical University School of Medicine Graduate School of Medicine, Department of Respiratory Medicine
REVIEW RETURNED	16-Feb-2022

GENERAL COMMENTS	This study used big data to examine the factors that influence exacerbations of COPD in the elderly in a community. The results of the study, such that income affects the frequency of exacerbations, are not novel, but the analysis using big data and propensity score matching is worthwhile. Major Comments 1 The paper is very difficult to understand due to the first appearance of several terms such as "65+" and "high number of exacerbations", which are not defined in the text. In addition, many of the statistical results are described as "related," which makes it difficult to know whether they are positive or negative. 2 The abstract states "The high number of consultations in primary care clinics was associated with a higher risk of COPD exacerbations (p=0.0687). " But I suppose this is because that the patients frequently visited physicians because of exacerbations. Rather I think the fact that there were many cases of exacerbations without prior consultations is more clinically important.
---

	minor 1 Table 1 is included in the supplementary material, and the text does not include table 1, but only table 2. The tables in the text should start with table1. 2 There is no information on smoking, which has a major impact on the pathogenesis and clinical course of COPD. 3 The presence of LTRA prescriptions suggests the presence of asthma complications. Concomitant asthma has a significant impact on exacerbation frequency. How about the results in the population excluding asthma diagnosis?
--	---

REVIEWER	Abraha, Iosief Azienda Unità Sanitaria Locale Umbria 2, Servizio Immunotrasfusionale
REVIEW RETURNED	28-Apr-2022

GENERAL COMMENTS	The authors intended to determine the prevalence of COPD in Poland and identify demographic, economic and environmental local community determinants of COPD exacerbations in elderly using Big Data approach. The proposed statistical approach is appropriate which is corroborated by additional method which the Propensity score method The paper however needs a major revision especially in terms of presentation of the data and analysis.  - It looks like that the population of interest were identified using the ICD-10 code. If so, outpatient COPD subject without icd-10 code might not be captured. There are several algorithms used to identify subject with COPD that were not necessarily admitted to hospital, e.g., using drug dispensation records (ATC code) (10.1186/s12913-019-4574-3). This should be clarified or acknowledged as a limitation at least in the discussion. - ICD-10 There is no mention in the article (or in the protocol) whether the ICD-10 codes were previously validated using an appropriate reference standard. This could be done using a small sample of medical charts. If this is not possible it must be acknowledged in the discussion. - The presentation of the results analysis is dense of information and it is hard follow. Presentation of the results should be provided in a fashioned way:  [ ] prevalence of the disease (confidence interval of the prevalence should also be calculated) [ ] descriptive characteristics of the determinants; [ ] analysis of the determinants; description of PSM analysis should follow directly the main analysis without creating separate paragraph for PSM. When presents the analysis of the determinants I suggest to provide Odds Ratios with confidence interval instead of P values - The same approach should be applied to the tables. Tables are hard to follow. Please provide separated tables for description and analysis; Also provide adequate heading in each table (for example Unadjusted** and Adjusted Logistic Models** column heading creates only confusion); Please provide exact P values (do not use NS); please provide with adequate explanations regarding drug codes R03AL and any other abbreviation used (N, T in the variable "did the patient die" where necessary. - Generally issues related to costs do have a different type of analysis of which I do not have any skill. - Abstract should be revised accordingly.
--

VERSION 1 – AUTHOR RESPONSE

Responses to Reviewers

Reviewer: 1

Dr. Syed Tariq, Luton and Dunstable University Hospital

Comments to the Author:

This study is a retrospective Big Data analysis to identify community-based factors associated with COPD exacerbations in the elderly population within a defined geographic area in Poland over a period of one year. After conversion of the data by applying propensity score matching, the study results are essentially negative, as apart from showing more exacerbations among the low income vs high income patients, there are no other significant results. The number of prescriptions/drugs, and the healthcare costs show statistical differences, but these are expected and likely reflect severity/GOLD class of COPD. All other community-based factors (specialty of primary care physician, forest cover, gender, number of PHC consultations, etc.) had little or no impact on rates of exacerbations. The association between the number of PHC consultations and hospital admissions may be explained simply by disease severity or even by an increased chance of being referred by the primary care physician to secondary care as these patients got seen more often by their primary care doctors.

R1_1. The study is retrospective, and such work has been done before. Hence, I suggest removing the word 'pioneering' from the first bullet point given in the 'Strengths and Limitations' section.

R1_2. There is no data on co-morbidities which could partly explain the differences in rates of exacerbation and hospitalization between the two study groups.

Answer to Reviewer 1:

Thank You for your assessment that the statistical methods are appropriate.

Thank You very much for any comments, remarks and suggestions to the Authors.

Answer R1_1:

Thank You very much for Your suggestion and comments to the Authors.

The risk factors for COPD and exacerbations are well known, while this study focused on local factors / determinants in a given patients living area in the Lodz province, and in this context it is a pioneering study especially in the local context. However we appreciate your comment and we reedited this part. (Please see 'Strengths and Limitations' section page no. 47 lines no. 10-18).

Answer R1_2:

Thank You for Your suggestion witch made description much clear. The variable "Codes of comorbidities with COPD" has 98.83% of missing data" of 472 314 of total 65+ patients, so it was excluded from the analysis.

(Please see Discussion section page no. 61 lines no. 42-50, and APPENDIX: 'Big Data Quality (Big Data Cleansing) Results' section page no. 84 lines no. 15-19).

R1_3. I do not understand the abbreviations given for the packages of reimbursed drugs (pp 12 & 14; Tables). These need a clear explanation.

Answer R1_3:

Thank You very much. The abbreviations given for the packages of reimbursed drugs were explained. (Please see Abbreviations section page no. 45 lines no. 3-45, Supplementary pages no. 74-75, and APPENDIX page no. 80 lines no. 19-29).

R1_4. The titles of tables 1 and 2 should be removed from the text. Both tables are too elaborate, and some of the data can be removed as it does not add to the analysis and results, such as the columns giving data on the total number of patients (n=16099) studied, and the mode (a descriptive statistic) for the number of PHC consultations.

Answer R1_4:

Thank You very much, as suggested, the titles of tables 1 and 2 and the data on the total number of patients (n = 16099) and the mode (descriptive statistics) for the "Number of consultations in PHC mode. Continuous var." have been removed from the text.

(Please see Results page no. 64-66 and APPENDIX Table 1A page 76 line 32, page 77 line 10 and lines 14-16).

R1_5. The breakdown of costs for the raw data and the data after PSM transformation deserves separate tables.

Answer R1_5:

Thank You for indicating all suggestions, they have all been taken into account.

The breakdown of costs for raw data and data after PSM transformation is shown in separate tables.

(Please see APPENDIX: Table 3A page 82 and 3B page 83).

There is no mileage in including the characteristics and specialties of primary care physicians in Table 1. These should simply be described in the text.

The specializations of primary care physicians in Table 1 are described in the text.

(Please see Results page no. 54 lines no. 42-48 and APPENDIX: Table 1A: page no. 78 lines 39-40 and 79 lines no. 3-7).

Similarly, the details of patients who had died should also be removed from Table 1 and be explained in the text.

Data on patients who died have been removed from table 1 and it was explained in the text.

(Please see Results page no. 53 lines no. 12-15 and APPENDIX: Table 1A: Page no. 79 lines no. 28-32).

The breakdown of age ranges at the end of Table 1 is not required and may simply be given in the text in 'Results' section.

Age ranges in Table 1A were removed, and they were given in the text in the "Results" section.

(Please see Results page no. 53 lines no. 51-59 and APPENDIX Table 1A page no. 80 lines no. 4-16).

The 'Missing data' given in each of the 7 rows under the heading of 'Drugs/Number of Packages' may be removed, and simply marked with a symbol with an explanation given at the end of Table 1.

"Missing data" have been removed and explained under the Table 1A.

(Please see APPENDIX: Table 1A: page no. 77 lines no. 28-38, page no. 78 lines no. 3-17).

R1_6. The first histogram in Appendix on Big Data Quality section is difficult to follow. Do the darker portions of the bars represent patients who had exacerbations? The histograms on gender are redundant and may be removed. The labels, etc. for the remaining two histograms need to be translated from Polish into English.

Answer R1_6:

Thank You very much for Your attention and comment. The histograms have been refined. The darker bars on the histograms before the application of the PSM technique, represent COPD patients with exacerbations (variable on the OX axis "Was patient hospitalized for J44 (Exacerbation)" 1 - yes, 0 - no). In the PSM technique, the selection of control for exacerbations took into account two variables: age and sex of the sample, we present both histograms. Labels for histograms had been translated. (Please see APPENDIX page no. 84line no. 31-57 and pages 85-88).

R1_7. Though the references are appropriate, they have lots of errors. The months and dates of publications are not required and may be removed. The sources of funding, and the descriptions in parentheses as to the type of publication (Review, Meta-analysis, Editorial, etc.) are not needed as well. Please check the accuracy of the references as there are several minor errors, and some of them are incomplete. Ref 19 contains an erroneous title.

Answer R1_7:

Thank You very much for this attention. References have been corrected. (Please see References section pages no. 67-69).

R1_8. The English needs improvement to make this manuscript more readable and acceptable. Although the statistical methods described are appropriate, I am unable to comment on the quality and outcomes of the analysis.

Answer R1_8:

Thank You very much for this attention. The English have been improved.

Reviewer: 2

Prof. Shigeo MURO, Nara Medical University School of Medicine Graduate School of Medicine
Comments to the Author:

This study used big data to examine the factors that influence exacerbations of COPD in the elderly in a community. The results of the study, such that income affects the frequency of exacerbations, are not novel, but the analysis using big data and propensity score matching is worthwhile.

Answer to Reviewer 2:

Thank You for your assessment that the analysis using Big Data and propensity score matching is worthwhile.

Thank you very much for suggestions and comments to the Authors.

From a literature review, risk factors for COPD exacerbations are well known for various cohorts.

In this study, we identified local determinants of exacerbations using the Big data datasets from Electronic Health Records of cohorts of COPD patients for a given area of central Poland (Lodz province). These local determinants of exacerbations make a new contribution to science as risk factors / protective factors of COPD exacerbations that have been identified for the area of Lodz province. (Please see Discussion section page no. 60 lines no. 26-43).

R2_Major Comments

R2_1_major The paper is very difficult to understand due to the first appearance of several terms such as "65+" and "high number of exacerbations", which are not defined in the text. In addition, many of the statistical results are described as "related," which makes it difficult to know whether they are positive or negative.

R2_2_major The abstract states "The high number of consultations in primary care clinics was associated with a higher risk of COPD exacerbations (p=0.0687). "

But I suppose this is because that the patients frequently visited physicians because of exacerbations. Rather I think the fact that there were many cases of exacerbations without prior consultations is more clinically important.

R2_minor

R2_1_minor Table 1 is included in the supplementary material, and the text does not include table 1, but only table 2. The tables in the text should start with table 1.

R2_2_minor There is no information on smoking, which has a major impact on the pathogenesis and clinical course of COPD.

R2_3_minor The presence of LTRA prescriptions suggests the presence of asthma complications. Concomitant asthma has a significant impact on exacerbation frequency. How about the results in the population excluding asthma diagnosis?

Reviewer: 3
Dr. Iosief Abraha, Azienda Unità Sanitaria Locale Umbria 2

Comments to the Author:

Answer major: R2_1:

Thank you very much for your suggestions and comments.

Indicated terms were defined in the text. The direction of statistical associations was shown in Results and Discussion section . (Please see Abbreviation page 45 lines 4-5, Results page 58 lines no. 33-39 and Discussion section page no. 60 lines no. 26-54).

Answer major: R2_2:

Thank you for this comment. We share your statement, that the patients frequently visit physicians because of exacerbations. We also share your comment on practical value of information on prior patient's consultation with the GP before hospitalization, however such information was not available from the sources we had access to.

Answer minor R2_1:

Thank you for your attention. Table number 2 in the text has been changed to number 1. Table numbering has been changed in line with your suggestion. (Please see Results section page no. 56 lines no. 10-20 and page 64 lines 3-5).

Answer minor R2_2:

Unfortunately this information was not available in the COPD patients' Electronic Health Records in the National Health Fund (NHF) database.

Answer minor R2_3:

Patients with code J44 were included in this study, without excluding codes for other diseases, including asthma, due to missing data in the variable "Codes for comorbidities with COPD (PHC)" has 98.83% of missing data of 472 314 total 65+ patients, so the variable was excluded from the analysis.

(Please see Discussion section page no. 61 lines no. 42-54, and APPENDIX: 'Big Data Quality (Big Data Cleansing) Results' section page no. 84 lines no. 16-18).

Answer to Reviewer 3:

Thank You for your assessment that the statistical methods are appropriate.

The authors intended to determine the prevalence of COPD in Poland and identify demographic, economic and environmental local community determinants of COPD exacerbations in elderly using Big Data approach.

The proposed statistical approach is appropriate which is corroborated by additional method which the Propensity score method

The paper however needs a major revision especially in terms of presentation of the data and analysis.

R3_1- It looks like that the population of interest were identified using the ICD-10 code. If so, outpatient COPD subject without icd-10 code might not be captured. There are several algorithms used to identify subject with COPD that were not necessarily admitted to hospital, e.g., using drug dispensation records (ATC code) (10.1186/s12913-019-4574-3). This should be clarified or acknowledged as a limitation at least in the discussion.

R3_2- ICD-10 There is no mention in the article (or in the protocol) whether the ICD-10 codes were previously validated using an appropriate reference standard. This could be done using a small sample of medical charts. If this is not possible it must be acknowledged in the discussion.

R3_3- The presentation of the results analysis is dense of information and it is hard to follow. Presentation of the results should be provided in a clear and concise way:

3a. * prevalence of the disease (confidence interval of the prevalence should also be calculated)

3b. * descriptive characteristics of the determinants;

3c. * analysis of the determinants; description of PSM analysis should follow directly the main analysis without creating a separate paragraph for PSM. When presenting the analysis of the determinants I suggest to provide Odds Ratios with confidence interval instead of P values

Thank You very much for your suggestion and comments.

Answer R3_1:

Thank You very much for your suggestion and comments.

Identification of patient population relied only on the ICD-10 J44 code in patients' primary care records, so COPD patients in specialist clinics and on hospital wards without this ICD-10 code might not be identified, what can be perceived as a study limitation.

Algorithms to identify patients with COPD who were not admitted to the hospital were not applied because data that would be related to prescribing drugs at a particular point of time (exacerbation) with the patient's admission to hospital were not available. This is a limitation of this study that has been added to the discussion. (Please see Discussion section page no. 61 lines no. 30-41).

Answer R3_2:

Thank You very much for your suggestion and comments.

In this Big Data study, J44 patient data was obtained from NFZ databases, not from primary care clinics so such validation was not possible.

It was elaborated in discussion sections. (Please see Discussion section page no. 61 lines no. 30-41).

Answer R3_3:

Thank you for your comments and comments to the Authors.

3a. the confidence intervals for the fractions (binomial distribution) for the populations were calculated and added. (Please see Results section page no. 53 lines no. 3-10 and APPENDIX page no. 79 lines no. 15-20).

3b. a descriptive characteristics of determinants were added in Supplementary materials section pages no. 74-75.

3c. the description of the PSM analysis has been moved according to your suggestion. (Please see Results section page no. 55 line no. 54).

Odds Ratios with confidence interval were presented in Table 1, according to Your suggestions.

(Please see Result section Table 1 and Table 2 pages no. 64-66, Supplementary materials pages 70-73, and APPENDIX pages 76-81).

R3_4- The same approach should be applied to the tables. Tables are hard to follow. Please provide separated tables for description and analysis; Also provide adequate heading in each table (for example Unadjusted** and Adjusted Logistic Models** column heading creates only confusion); Please provide exact P values (do not use NS); please provide with adequate explanations regarding drug codes R03AL and any other abbreviation used (N, T in the variable "did the patient die" where necessary).

Answer R3_4:

Thank you very much for these comments which make this manuscript much more clear.

The separated tables for description and analysis were provided as Table 1 and Table 2 in the body of manuscript.

The tables have been changed according to your suggestions: headings, p-value, and explanations regarding drug codes and other abbreviation.

(Please see Results section pages no. 64-66, and Abbreviations page 45 lines no. 3-45).

R3_5- Generally issues related to costs do have a different type of analysis of which I do not have any skill.

Answer R3_5:

Thank You very much for your suggestion and comments.

In this study the patients' socioeconomic status was indicated as "Total personal income of residents per PHC post-codes (Tax Office)" were divided into 3 categories, and was included into logistic regression models as categorical variable.

R3_6- Abstract should be revised accordingly.

Answer R3_6:

Thank You very much for your suggestion and comments.

The abstract has been revised accordingly in line with your suggestions and comments.

(Please see Abstract section page no. 45 lines no. 47-60 and page no. 46 lines no. 3-47).

VERSION 2 – REVIEW

REVIEWER	Tariq, Syed Luton and Dunstable University Hospital, Respiratory Medicine
REVIEW RETURNED	18-Jul-2022

GENERAL COMMENTS	ange 'cites' to 'cities'
--------------------------

	Study size/Power Calculation, last line: change 'Total of' to 'A total of'; and change 'allow to 80%' to 'allow 80%' Page 54, line 3: change 'per COPD 65 patient.' to 'per 65+ COPD patient.' Page 54, line 8: change 'four and more' to 'four or more' Page 54, line 21: change '(min. 25, max. 88)' to '(range 25 - 88)', and remove 'old' Page 55, line 9: change '(11.84%) of missing data)' to '(11.84% missing data)' Page 56: remove the title of Table 1 from the text Page 56, 2nd para, 1st line: change 'the similar relationships' to 'similar relationships' Page 56, 2nd para, line 2: Re-write this sentence - Median age was the same for patients with and without exacerbations (75 years, q1=69, q3=81, for both groups). Page 56, 3rd para: Re-write as 'Patient consultations, and consultations for COPD in PHC clinics had a weak association with exacerbations (high vs low p = 0.0687). Page 57, line 2: change 'packaging' to 'packages'; change 'The patients with exacerbations realized' to 'Patients with exacerbations had' Page 57, line 7: change 'reinvestment' to 'reimbursement' Page 57, line 14: change 'There were no association' to 'There was no association' Page 58, line 4: Re-write as 'Two variables seemed important:' Page 58, line 6: change 'insignificantly lower' to 'a slightly lower' Page 58, line 10: Re-write this sentence - COPD patients with a high number of consultations (>12) in PHC showed a trend for more exacerbations compared to those with a low number (<5) of consultations (OR 1.261, CI 0.974-1.633). Page 59, line 2: change 'can be increased as' to 'may increase as' Page 59, line 7: remove 'precisely' Page 59, line 11: change 'on the disease itself' to 'on disease prevalence' Page 59, line 20: change 'However, that tendency was not found in our study where 59.38% of total COPD' to 'However, in our study 59.38% of COPD' Page 60, 2nd para, line 1: change 'and high income' to 'and a low income' Page 60, 2nd para, line 2: change 'were also significant determinants' to 'were important determinants' Page 60, 2nd para, line 3: Re-write 'controlled for....' to 'whereas there was no effect of specialization of the PHC physician, forest cover of the local area, and the location of the gmina on rates of exacerbations.' Page 60, 3rd para, line 1: change 'have more than 1.26 times the chances of' to 'had a slightly higher risk of' Page 60, 3rd para, line 2: remove 'but this was on the border of statistical significance' Page 61, line 2: change 'on the COPD exacerbation' to 'on COPD exacerbations' Page 61 line 5: change 'This disparity may be related to increased' to 'This disparity may be related partly to increased' Page 61, line 6: change 'agricultural exposure' to 'agricultural dust/aerosol exposure', and remove 'among others' Page 61, line 7: change 'suggest this may be associated' to 'suggest an association', and change 'indoor air exposure associated with' to 'indoor pollution due to' Page 62, 2nd sentence: change 'may be also used' to 'may also be used'
--	---

	Page 62, Conclusions, last sentence: change 'However, further study' to 'Further study' Table 1 title: change 'cohort of COPD elderly patients' to 'cohort of elderly COPD patients' Table 1, 1st column, page 1: Change the heading 'Healthcare uses' to 'Healthcare use' Table 1, page 2: The authors have categorised the number of COPD consultations into narrow bands of 0, 1, 2, 3 and 4 or more. This is of little value. To simplify, a cut off of 2 or more vs 0-1 consultations may work better. I will be happy to look at the 2nd revision.
--	---

REVIEWER	MURO, Shigeo Nara Medical University School of Medicine Graduate School of Medicine, Department of Respiratory Medicin
REVIEW RETURNED	23-Jun-2022

GENERAL COMMENTS	none
------

VERSION 2 – AUTHOR RESPONSE

Reviewer: 2

Prof. Shigeo MURO, Nara Medical University School of Medicine Graduate School of Medicine

Comments to the Author: none

Answer to Reviewer: 2

Thank you very much.

Reviewer: 1

Dr. Syed Tariq, Luton and Dunstable University Hospital

Comments to the Author:

The revised manuscript provides a more balanced view. The authors have adequately addressed the concerns that I, and my co-reviewers had raised. They have also corrected and improved the list of references. The results of this study are essentially negative as, after propensity score matching, apart from low vs high income, all other socio-economic and environmental factors were statistically non-significant for COPD exacerbations. There is only a trend for a high number of consultations at PHCs with more exacerbations ($p = 0.0687$).

The association of low income with increased risk of exacerbations can be explained by a number of other co-factors. For example, patients with a low income could simply be having more severe COPD, or a higher burden of co-morbidities, or they are less compliant with their regular inhalers/medications. They may also have a less healthy lifestyle, or live in poor and over-crowded conditions. These points are worth mentioning in the discussion.

Despite mostly negative results, the study has value as it reveals the prevalence of COPD in a defined area of Poland and highlights that there is little impact of place of residence, local forest cover, or consultations with a family physician vs a more specialised physician, within this area.

Answer to Reviewer: 1

Thank you very much for all corrections, comments and suggestions that are very helpful in improving our manuscript.

We accepted and implemented all your amendments in accordance with your comments and suggestions.

Co-factors that may contribute to the association of low income with increased risk of exacerbations have been added as suggested. (Please see Discussion section). Page no. 68 lines no. 5-15.

Reviewer: 1

There are a few typos requiring correction, and some statements needing improvement:

Answer to Reviewer: 1

Thank you very much for all corrections, comments and suggestions that are very helpful in improving our manuscript.

We accepted and implemented all your corrections and suggestions.

Reviewer: 1

Intro, last line: change 'cites' to 'cities'

Answer to Reviewer: 1

Please see Introduction section: page no. 51 line no. 47.

Reviewer: 1

Study size/Power Calculation, last line: change 'Total of' to 'A total of'; and change 'allow to 80%' to 'allow 80%'

Answer to Reviewer: 1

Please see Study size/Power Calculation section: page no. 56 line no. 10.

Reviewer: 1

Page 54, line 3: change 'per COPD 65 patient.' to 'per 65+ COPD patient.'

Answer to Reviewer: 1

Please see Results section: page no. 57 lines no. 31-33.

Reviewer: 1

Page 54, line 8: change 'four and more' to 'four or more'

Answer to Reviewer: 1

Please see Results section: page no. 57 line no. 44.

Reviewer: 1

Page 54, line 21: change '(min. 25, max. 88)' to '(range 25 - 88)', and remove 'old'

Answer to Reviewer: 1

Please see Results section: page no. 58 line no. 19.

Reviewer: 1

Page 55, line 9: change '(11.84%) of missing data)' to '(11.84% missing data)'

Answer to Reviewer: 1

Please see Results section: page no. 58 line no. 33.

Reviewer: 1

Page 56: remove the title of Table 1 from the text

Answer to Reviewer: 1

Please see Results section: page no. 59 line no. 33.

Reviewer: 1

Page 56, 2nd para, 1st line: change 'the similar relationships' to 'similar relationships'

Answer to Reviewer: 1

Please see Results section: page no. 62 line no. 5.

Reviewer: 1

Page 56, 2nd para, line 2: Re-write this sentence - Median age was the same for patients with and without exacerbations (75 years, q1=69, q3=81, for both groups).

Answer to Reviewer: 1

Please see Results section: page no. 62 lines no. 7-13.

Reviewer: 1

Page 56, 3rd para: Re-write as 'Patient consultations, and consultations for COPD in PHC clinics had a weak association with exacerbations (high vs low $p = 0.0687$).

Answer to Reviewer: 1

Please see Results section: page no. 62 lines no. 14-22.

Reviewer: 1

Page 57, line 2: change 'packaging' to 'packages'; change 'The patients with exacerbations realized' to 'Patients with exacerbations had'

Answer to Reviewer: 1

Please see Results section: page no. 62 line no. 47.

Reviewer: 1

Page 57, line 7: change 'reinvestment' to 'reimbursement'

Answer to Reviewer: 1

Please see Results section: page no. 63 line no. 5.

Reviewer: 1

Page 57, line 14: change 'There were no association' to 'There was no association'

Answer to Reviewer: 1

Please see Results section: page no. 63 line no. 22.

Reviewer: 1

Page 58, line 4: Re-write as 'Two variables seemed important:'

Answer to Reviewer: 1

Please see Results section: page no. 65 line no. 12.

Reviewer: 1

Page 58, line 6: change 'insignificantly lower' to 'a slightly lower'

Answer to Reviewer: 1

Please see Results section: page no. 65 lines no. 17-19.

Reviewer: 1

Page 58, line 10: Re-write this sentence - COPD patients with a high number of consultations (>12) in PHC showed a trend for more exacerbations compared to those with a low number (<5) of consultations (OR 1.261, CI 0.974-1.633).

Answer to Reviewer: 1

Please see Results section: page no. 65 lines no. 23-36.

Reviewer: 1

Page 59, line 2: change 'can be increased as' to 'may increase as'

Answer to Reviewer: 1

Please see Discussion section: page no. 66 line no. 8.

Reviewer: 1

Page 59, line 7: remove 'precisely'

Answer to Reviewer: 1

Please see Discussion section: page no. 66 line no. 19.

Reviewer: 1

Page 59, line 11: change 'on the disease itself' to 'on disease prevalence'

Answer to Reviewer: 1

Please see Discussion section: page no. 66 line no. 28.

Reviewer: 1

Page 59, line 20: change 'However, that tendency was not found in our study where 59.38% of total COPD' to 'However, in our study 59.38% of COPD'

Answer to Reviewer: 1

Please see Discussion section: page no. 66 line no. 49.

Reviewer: 1

Page 60, 2nd para, line 1: change 'and high income' to 'and a low income'

Answer to Reviewer: 1

Please see Discussion section: page no. 67 line no. 19.

Reviewer: 1

Page 60, 2nd para, line 2: change 'were also significant determinants' to 'were important determinants'

Answer to Reviewer: 1

Please see Discussion section: page no. 67 line no. 21.

Reviewer: 1

Page 60, 2nd para, line 3: Re-write 'controlled for....' to 'whereas there was no effect of specialization of the PHC physician, forest cover of the local area, and the location of the gmina on rates of exacerbations.'

Answer to Reviewer: 1

Please see Discussion section: page no. 67 lines no. 26-34.

Reviewer: 1

Page 60, 3rd para, line 1: change 'have more than 1.26 times the chances of' to 'had a slightly higher risk of'

Answer to Reviewer: 1

Please see Discussion section: page no. 67 lines no. 38-40.

Reviewer: 1

Page 60, 3rd para, line 2: remove 'but this was on the border of statistical significance'

Answer to Reviewer: 1

Please see Discussion section: page no. 67 line no. 42.

Reviewer: 1

Page 61, line 2: change 'on the COPD exacerbation' to 'on COPD exacerbations'

Answer to Reviewer: 1

Please see Discussion section: page no. 68 line no. 28.

Reviewer: 1

Page 61 line 5: change 'This disparity may be related to increased' to 'This disparity may be related partly to increased'

Answer to Reviewer: 1

Please see Discussion section: page no. 68 line no. 35.

Reviewer: 1

Page 61, line 6: change 'agricultural exposure' to 'agricultural dust/aerosol exposure', and remove 'among others'

Answer to Reviewer: 1

Please see Discussion section: page no. 68 line no. 37.

Reviewer: 1

Page 61, line 7: change 'suggest this may be associated' to 'suggest an association', and change 'indoor air exposure associated with' to 'indoor pollution due to'

Answer to Reviewer: 1

Please see Discussion section: page no. 68 lines no. 40-45.

Reviewer: 1

Page 62, 2nd sentence: change 'may be also used' to 'may also be used'

Answer to Reviewer: 1

Please see Conclusions section: page no. 69 line no. 35.

Reviewer: 1

Page 62, Conclusions, last sentence: change 'However, further study' to 'Further study'

Answer to Reviewer: 1

Please see Conclusions section: page no. 69 lines no. 40-43.

Reviewer: 1

Table 1 title: change 'cohort of COPD elderly patients' to 'cohort of elderly COPD patients'

Answer to Reviewer: 1

Please see Table 1 title: page no. 59 lines no. 42 and lines 46-49.

Reviewer: 1

Table 1, 1st column, page 1: Change the heading 'Healthcare uses' to 'Healthcare use'

Answer to Reviewer: 1

Please see Table 1, 1st column, page 1: page no. 60 line no. 33.

Reviewer: 1

Table 1, page 2: The authors have categorised the number of COPD consultations into narrow bands of 0, 1, 2, 3 and 4 or more. This is of little value. To simplify, a cut off of 2 or more vs 0-1 consultations may work better.

Answer to Reviewer: 1

Variable "Number of COPD consultations" in Table 1, page 2 has been categorized into 2 categories: 0-1 consultations vs. 2 or more consultations.

Please see Table 1, page 2: page no. 61 lines no. 13-17.

Reviewer: 1

I will be happy to look at the 2nd revision.